# Drug Combinations Targeting FAK and MEK Overcomes Tumor Heterogeneity in Glioblastoma

**DOI:** 10.3390/pharmaceutics17050549

**Published:** 2025-04-23

**Authors:** Muhammad Furqan, Richard J. R. Elliott, Peter W. K. Nagle, John C. Dawson, Roza Masalmeh, Virginia Alvarez Garcia, Alison F. Munro, Camilla Drake, Gillian M. Morrison, Steven M. Pollard, Daniel Ebner, Valerie G. Brunton, Margaret C. Frame, Neil O. Carragher

**Affiliations:** 1Edinburgh Cancer Research, Institute of Genetics and Cancer, Western General Hospital, University of Edinburgh, Edinburgh EH4 2XU, UK; mfurqan@exseed.ed.ac.uk (M.F.); richard.elliott1@ed.ac.uk (R.J.R.E.); peter.nagle@ed.ac.uk (P.W.K.N.); john.c.dawson@ed.ac.uk (J.C.D.); roza.masalmeh@ed.ac.uk (R.M.); virginia.alvarezgarcia@ed.ac.uk (V.A.G.); alison.munro@ed.ac.uk (A.F.M.); v.brunton@ed.ac.uk (V.G.B.); m.frame@ed.ac.uk (M.C.F.); 2Cancer Research UK Scotland Centre, University of Edinburgh, Edinburgh EH4 2XR, UK; gillian.morrison@ed.ac.uk (G.M.M.); steven.pollard@ed.ac.uk (S.M.P.); 3Centre for Regenerative Medicine, Institute of Regeneration and Repair, University of Edinburgh, Edinburgh EH16 4UU, UK; 4Nuffield Department of Medicine, University of Oxford, Old Road Campus, Oxford OX3 7BN, UK; daniel.ebner@ndm.ox.ac.uk

**Keywords:** focal adhesion kinase (FAK), MEK, glioblastoma, tumor heterogeneity, combination therapy, chemogenomic screening

## Abstract

**Background/Objectives**: Glioblastoma (GBM) is an aggressive brain tumor with limited treatment options and poor prognosis, largely owing to its heterogeneity and the involvement of multiple intracellular signaling pathways that contribute to drug resistance. While recent advancements in targeted drug combination therapies, such as dabrafenib and trametinib, show promise for certain GBM subgroups, identifying effective drug combinations across the broader GBM population remains a challenge. Integrin-mediated signaling, particularly through Focal Adhesion Kinase (FAK), plays a pivotal role in GBM pathogenesis and invasion, making it a potential therapeutic target and component of future drug combination strategies. **Methods**: In this study, we utilized a chemogenomic screening approach to identify synergistic drug combinations that target FAK in glioblastoma. We initially employed a CRISPR-engineered GBM model to assess the effects of FAK depletion and subsequently discovered that combining FAK inhibitors such as VS4718 with MEK inhibitors, particularly trametinib, demonstrated synergistic effects. This potent combination was validated using various 2D and 3D assays, including cell viability/apoptosis assessment, synergistic analysis, cellular imaging, and target engagement assays. This combination also effectively inhibited spheroid growth and invasion across a diverse panel of patient-derived GBM stem cells. Molecular mechanisms underlying these effects include suppression of multiple kinase signaling pathways and enhanced apoptosis, elucidated using Reverse-Phase Protein Array (RPPA) profiling and Western blot validation. **Result**: In vivo, combination therapy significantly reduced the tumor volume in orthotopic transplantation models. **Conclusions**: These findings suggest that the combination of FAK and MEK inhibitors represents a promising therapeutic strategy to overcome the challenges of GBM treatment.

## 1. Introduction

Glioblastoma (GBM) remains a cancer of high unmet clinical need, despite considerable efforts to develop new treatments. The current standard of care for newly diagnosed GBM patients consists of surgery, temozolomide (TMZ), and ionizing radiation (IR), which provides an overall 15-month survival benefit and has not been improved since the EORTC-NCIC trial was published in 2005 [1]. The self-renewing capacity of GBM stem cells is an inherent feature of GBM to which the tumor microenvironment contributes supporting adaptive signal rewiring, heterogeneous tumor evolution, and therapeutic resistance [2,3]. Current treatments and clinical trials of modern targeted therapies have been largely ineffective because they fail to address inherent tumor heterogeneity, contributing to incomplete treatment response and resistance due to the emergence of hypermutator phenotypes and rapid rewiring across multiple signaling pathways [4,5].

Large-scale genomic analyses performed over the past decade have enhanced our understanding of GBM biology, supporting the classification of GBM into different subtypes based on gene signatures that may help predict prognosis and therapeutic response [6,7,8]. However, this new understanding of the GBM genomic landscape has not yet yielded effective personalized medicine strategies across the GBM population, predominantly as a result of the fact that GBM is driven by multiple intracellular signaling pathways rather than single gene alterations. Thus, the challenge in GBM is identifying effective drug combinations that overcome tumor heterogeneity and therapeutic resistance by targeting the tumor’s ability to rewire its signaling networks following treatment with single-pathway blocking therapy [4].

Despite these challenges, new hope has been provided by the recent Food and Drug Administration (FDA) approval of the combination of the targeted BRAF inhibitor dabrafenib and MEK inhibitor trametinib for the treatment of nearly any type of advanced solid tumor with a BRAF^V600E^ mutation [9]. Studies have shown that treatment of tumors with BRAF inhibitors (such as dabrafenib) results in the development of resistance by activating signals from the MEK protein [10]. Thus, combining dabrafenib with the MEK inhibitor trametinib was postulated to prevent tumors from using this escape mechanism and has demonstrated increased efficacy in preclinical tumor models and clinical studies [10,11,12]. The phase II Rare Oncology Agnostic Research (ROAR) clinical trial of dabrafenib plus trametinib in patients with BRAF V600E mutation-positive recurrent or refractory high-grade glioma (including GBM) and low-grade glioma constitutes part of a larger basket trial BRF117019 (NCT02034110), evaluating dabrafenib and trametinib combination therapy across multiple BRAF V600E-mutant tumors. The ROAR clinical trial’s primary endpoint of investigator-assessed overall response rate was 33% in the high-grade glioma cohort. Secondary endpoints of the median duration of response were 31.2%, progression-free survival was 5.5%, and median overall survival was 17.6% in high-grade glioma [13]. While only 2% of adult GBM express BRAF V600E, the incidence of activating oncogenic BRAFV600E mutation is higher across other central nervous system primary tumors in pediatric and adult patients [14]; thus, dabrafenib combined with trametinib represents a potential treatment option for these subsets of CNS tumor patients.

With an almost infinite multidrug possibility, identifying the most effective drug combinations across other GBM patient subgroups presents a significant challenge, as evidenced by the failure of the majority of drug combination trials performed in GBM to date [15]. The scientific rationale and preclinical evidence supporting many drug combination trials are often unclear, and it appears that many drug combination strategies are opportunistic with selection based on pragmatic considerations, including drug access, and often representing an after-thought to rescue the poor response to single-agent therapy. Recent high-throughput systematic testing of drug combinations across cancer cell models indicates that synergy between drugs is rare and highly context-dependent [16]. Thus, the application of more evidence-based approaches to the discovery of novel drug combination strategies incorporating GBM models that represent the heterogeneity of disease is required to validate and prioritize the most promising drug combination strategies for clinical development.

Altered extracellular matrix (ECM) remodeling and intracellular signaling mediated by the integrin family of ECM receptors and their downstream kinase effectors, such as Focal Adhesion Kinase (FAK), have been proposed as important mechanisms influencing GBM pathogenesis, formation of the tumor niche, and brain tissue invasion [17]. Overexpression of integrin receptor subunits β1, β3, α3, α5, and αv, and high expression of FAK and its closely related homolog PYK2, are associated with a reduced overall survival of GBM patients [18,19,20,21]. However, therapeutic strategies to directly inhibit aberrant integrin-mediated ECM adhesion and signaling, such as the peptidomimetic drug candidate cilengitide, which blocks the ECM-RGD binding site of αvβ3 and αvβ5 integrins, failed to improve the overall survival of patients with GBM despite promising preclinical data [22,23]. The failure of such integrin-targeting therapy has been proposed to be a consequence of a variety of pro-survival bypass signaling pathways mediated by multiple integrin receptors [17] and other upstream ECM adhesion signaling receptors known to be dysregulated in GBM, including discoidin domain receptors (DDR) [24] and CD44 [25]. These findings suggest that targeting downstream mediators positioned at the intersection of multiple integrins and receptor tyrosine kinase signaling pathways, such as FAK, may have additional therapeutic benefits. We and others have demonstrated that FAK is an important downstream mediator of the invasion and survival pathways in multiple cancer cell types, including GBM [26,27,28]. In human GBM cell models, both small-molecule pharmacological inhibition and genetic knockdown of FAK suppressed the proliferation, survival, and 3D invasion of glioma stem cells [29,30]. Pharmacological inhibition of FAK has also been shown to enhance the effect of temozolomide on tumor growth in a C57BL/6-GL261 mouse glioma model [31] and displays radiosensitizing effects in a subset of GBM cell models [29]. Furthermore, in a murine in vivo surgical resection model, FAK and PYK2 signaling was upregulated in recurrent tumor samples [21]. Treatment with the small molecule PYK2/FAK inhibitor PF-562271 reversed PYK2/FAK signaling activation in recurrent tumors, reducing tumor volume by 43% and increasing animal survival by 33% [21]. Although these results are promising, more effective and durable responses are desirable. Collectively, these studies indicate that FAK is a promising therapeutic target in GBM but may only yield maximum patient benefit when exploited as part of a synergistic therapeutic drug combination.

In this study, we employed an unbiased chemogenomic screening approach in an isogenic model of FAK-kinase-deficient (KD) and FAK-wild-type (WT) expressing GBM cells to identify drug combinations that could potentially synergize with the depletion of FAK kinase activity. Our screen identified a number of drug candidates, including four distinct MEK inhibitors that enhanced GBM cell death when combined with FAK-KD cells, relative to FAK-WT. Focusing on the most clinically advanced MEK inhibitor, trametinib, we demonstrated low-level synergistic activity upon FAK-WT GBM cell survival and invasion endpoints when combined with multiple small-molecule FAK inhibitors. Post-translational pathway profiling indicated that trametinib promotes multiple compensatory signaling and pathway rewiring events in GBM cells when used as a single agent. Validation studies across a heterogeneous panel of patient-derived GBM stem cells demonstrated potent FAK+MEK inhibitor combination activity at nanomolar concentrations in 2D and 3D phenotypic assays and reduced tumor area in an in vivo orthotopic transplantation model. Our systematic GBM phenotypic discovery platform demonstrates the robustness of the FAK+MEK inhibitor combination treatment across heterogeneous GBM stem cell models, supporting the case for further preclinical development in clinical studies.

## 2. Material and Methods

### 2.1. Cell Culture

To explore the role of FAK in GBM, we utilized a recently described transformed neural stem cell-derived GBM cell model featuring CRISPR/Cas9-mediated deletion of the tumor suppressor genes *Nf1* (N) and *Pten* (P), and concomitant expression of the constitutively active oncogenic Epidermal growth factor receptor (EGFR (E)) variant EGFRVIII (hereafter termed NPE cells) [32]. To investigate the function of FAK in NPE cells, we performed CRISPR/Cas9-mediated knockout of FAK in NPE cells and re-expressed FAK^K454R^ kinase deficient mutant protein (NPE-FAK-Kinase-deficient or -KD)) and wild-type FAK protein at endogenous levels (NPE-FAK-WT) in the same clonal population (see Appendix A), thus creating paired isogenic NPE cells whose phenotypic differences can be attributed solely to the expression of catalytically active FAK.

The patient-derived glioblastoma stem cell lines used in this study were obtained from the Glioma Cellular Genetics Resource, Edinburgh (Appendix A). Cells were maintained in DMEM/HAMS-F12 media (Merck Life Sciences, Gillingham, UK, #D8437) supplemented with glucose (8 mM, Merck Life Sciences, Gillingham, UK, #G8644), beta-mercaptoethanol (100 µM, Thermo Fisher Scientific, Paisley, Scotland, UK, #31350-010), MEM non-essential amino acids (Thermo Fisher Scientific, Paisley, Scotland, UK, #11140-035), BSA (0.015%, Thermo Fisher Scientific, Paisley, Scotland, UK, #15260-037), recombinant hFGF basic (12 ng/mL, Peprotech, Thermo Fisher Scientific, Paisley, Scotland, UK, #100-18b), and recombinant mEGF (12 ng/mL, Peprotech, Thermo Fisher Scientific, Paisley, Scotland, UK, #315-09) as well as laminin (4–10 ug/mL, R&D Systems, Minneapolis, MN, USA, #3446-0050-01), B27 (1:100, Thermo Fisher Scientific, Paisley, Scotland, UK, #17504-044) and N2 supplements (1:200, Thermo Fisher Scientific, Paisley, Scotland, UK, #17502-048) [33]. Cells were plated onto 384-well microplates (Greiner Bio-One, Stonehouse, UK, #781091) after pre-coating with laminin (10 µg/mL in media, R&D Systems, Minneapolis, MN, USA, #3446-005-01), at 500–1500 cells/well.

### 2.2. Compound and Drug Combination Screening

Kinase Chemogenomic Set (KCGSv1.0, 187 compounds) was provided by the Structural Genomics Consortium (SGC-UNC, Chapel Hill, NC, USA). The Library of Pharmacologically active compounds (LOPAC, 1280 compounds, #LO4100) was obtained from Merck Life Sciences (Gillingham, UK). The TargetMol anti-cancer set (330 compounds, #L2100) was purchased from TargetMol Chemicals Inc. (Wellesley Hills, MA, USA). The custom C3L library (789 compounds) [34] was directly supplied by our partners at the University of Oxford (Target Discovery Institute, Oxford, UK). Compound plates (10 mM starting conc., 1.5 uL/well) were distributed as ‘assay ready’ copies in 384 w format using a BioMek FX liquid handling platform (Beckman Coulter Life Sciences, High Wycombe, UK) and were initially diluted in media (1:50 dilution, 75 µL/well final volume, 200 µM, 2% DMSO (*v*/*v*)) using a ViaFill liquid reagent dispenser (Integra Biosciences Pty Ltd, Berkshire, UK). Staurosporine (1 µM, Merck Life Sciences, Gillingham, UK, #S5921) was added to column 1 of each compound plate as a positive control for cell death (*n* = 16 per plate). The plates were transferred to the BioMek FX platform and serially diluted to 20*x* final concentration required, in media containing 2% DMSO (*v*/*v*). Finally, 2.5 µL of each diluted compound was transferred to the cell plates (1:20 dilution, final volume 50 µL, 0.1% DMSO (*v*/*v*) in all wells, including ‘untreated controls’ *n* = 48 per plate) at final compound concentrations of 1000, 300, 100 and 30 nM. The plates were incubated at 37 °C and 5% CO_2_ for 48 h before the viability assay (PrestoBlue, Thermo Fisher Scientific, Paisley, Scotland, UK, #A13262). Fluorescence background counts (from ‘cell free’ wells) were subtracted from the raw data and the data was processed via univariate analysis using TIBCO^®^ Spotfire® Analyst (version 12.0.3 LTS) with PerkinElmer Signals VitroVivo 3.1.0—High Content Profiler App (Revvity Signals Software, Inc. Waltham, MA, USA).

Hit compounds were validated in 384w format in triplicate by an 8-point dose–response (1000 to 10 nM). Dose–response profiles and IC_50_ ranges were used to build drug combination dose ratio matrices (384w format, 7 × 7 matrix per combination pair) using a D300e digital dispenser (Tecan.com). After 48 h, cell viability was quantified (PrestoBlue, Thermo Fisher Scientific, Paisley, Scotland, UK, #A13262) and the data were normalized to internal DMSO controls (% Cell Survival) after background correction.

### 2.3. Cell Viability Assay

NPE-FAK-WT and FAK-KD cells were seeded in 384-well plates at 400 cells/well and incubated for 24 h before treatment. Cells were incubated with compounds for 48 h, untreated cells were incubated with 0.1% DMSO as vehicle control (*n* = 48 per plate). PrestoBlue HS Cell viability reagent (5 µL, Thermo Fisher Scientific, Paisley, Scotland, UK, #A13262) was added to each well using a ViaFill dispenser, and the plates were incubated for 2 h at 37 °C. Fluorescence emission was measured on an EnVision 2101 multilabel plate reader (Revvity, Waltham, MA, USA). Background fluorescence was subtracted from ‘cell free’ wells. All conditions were then normalized on a plate-by-plate basis to internal DMSO controls (average of *n* = 48) to provide % survival and Z-score quantification to compare relative survival (NPE-FAK-KD vs. NPE-FAK-WT) and induced drug sensitivity in a PTK-suppressed background. Dose–response validations were carried out in triplicate with a minimum of 3 biological replicates.

### 2.4. Cellular Imaging and Quantification

Human glioma stem cells (GCGR cell lines) in 384w format were quantified at assay end points by nuclei staining and imaging using an ImageXpress high-content microscope (Molecular Devices LLC, San Jose, CA, USA). Cells were fixed in situ by the addition of 8% formaldehyde (equivalent volume, cooled plates) using a multidrop dispenser (Thermo Fisher Scientific, Paisley, Scotland, UK, #5840330), incubated for 30 min, and fixative removed using BioTek multi-well plate washer (Agilent, Sanat Clara, CA, USA). The cells were then stained with Hoechst-33342 (Thermo Fisher Scientific, Paisley, Scotland, UK, #H1399, 2 ug/mL in PBS, 0.1% Triton-X100 (*v*/*v*)) at RT for 20 min, washed, and sealed with foil. Processed plates were then loaded into an ImageXpress-XL microscope using a KX robotic arm (Peak Analysis and Automation, Farnborough, UK) and imaged with a 20× objective (six sites per well). Total Nuclei counts were quantified using the MetaXpress built-in analysis module (Count Nuclei) (Molecular Devices). Experiments were re-confirmed over at least 3 biological replicates. For representative images, fixed cells were stained with Hoechst (1:5000, 2 ug/mL), Phalloidin-594 (ab176757, Abcam, Cambridge, UK 1:1500), and Concanvalin A (Thermo Fisher Scientific, Paisley, Scotland, UK, #C11252, 30 ug/mL). 

### 2.5. Quantification of Tumor Area in IHC Stained Sections

Tumor areas in immunohistochemically stained sections were quantified using ImageJ software (version 1.54i). High-resolution images of stained tissues were first converted to grayscale. Thresholds were adjusted to isolate positive staining and to distinguish it from the background. The software’s measurement tools were then used to calculate the area of the stained tumor regions, which was expressed as the total area of the stained tissue section. This approach allowed for consistent and objective quantification of the tumor burden across different samples.

### 2.6. Synergy Calculation

Experiments were carried out in 384w format with multiple 7 × 7 dose combination matrices (including ‘drug alone’ arms) in triplicate. Typical dose ranges for the compounds used were VS4718, AMP-945 (500–100 nM in 2D and 3 µM in 3D), and trametinib, GDC0623 (25-1 nM in 2D and 250 nM in 3D). At the end of the assay (72 h for human glioma cells and 48 h for NPE-derived lines), cells were fixed and stained with Hoechst (as described above). Image quantification of nuclei counts, followed by DMSO normalization, yielded % cell survival measurements, which were then analyzed using SynergyFinder software (version 3.0, https://synergyfinder.org/) [35], incorporating multiple synergy models including Bliss, Loewe, HSA, and ZIP synergy models for robust synergistic ranking with a combined sensitivity score (CSS) [36,37,38]. Dose combinations were reproduced over a minimum of 3 biological replicates.

### 2.7. Three-Dimensional Spheroid Invasion Assay

Cells were seeded in 96-well ultra-low-attachment “U” bottom plates (Corning Life Sciences, NY, USA #7007) at 2000 cells per well in normal growth medium and centrifuged at 200× *g*. Cells were allowed to form a single spheroid per well over 3–4 days before being overlaid with Matrigel (Thermo Fisher Scientific, Paisley, Scotland, UK, #11543620, 4 mg/mL). Subsequently, they were treated with drugs with continuous monitoring of spheroid outgrowth every 3 h over 12 days using an IncuCyte S3 live cell imaging system (Satorius, Gottingen, Germany) and analyzed using the built-in 3D Spheroid Invasion module. The imaging channels ’Phase + Brightfield’ with a 10× objective were used to observe spheroid size and invasion. Invasion assays were repeated over 3 biological replicates.

### 2.8. Western Blotting

Cells or spheroids were washed with PBS and lysed in ice cold RIPA buffer (50 mM Tris-HCl pH 8, 0.1% sodium dodecyl sulfate, 0.5% sodium deoxycholate, 1% Triton X-100, 150 mM NaCl) supplemented with PhosSTOP™ Phosphatase Inhibitor (Roche Diagnoistic Ltd., West Sussex, UK, #4906845001) and cOmplete™ ULTRA Protease Inhibitor (Roche, #5892953001) cocktail for half an hour on ice with gentle agitation. Lysates were centrifuged at high speed to clear the debris, and the supernatants were quantified using Coomassie Plus Protein Assay Reagent (Thermo Fisher Scientific, Paisley, Scotland, UK, #1856210), followed by protein normalization. Then, samples were resolved in 4–15% Mini-PROTEAN TGXTM gels, transferred onto PVDF membranes (Bio-Rad Laboratories, Watford, UK, #1704156), and blocked in Roche block followed by incubation with primary antibodies overnight. The next day, the membranes were washed with PBS-T and incubated with HRP-linked secondary antibodies for 1 h. The membranes were again washed three times with PBS-T and visualized using Clarity Western ECL substrate (Bio-Rad Laboratories, Watford, UK, ##1705061) on a ChemiDoc imaging system (Bio-Rad Laboratories, Watford, UK). The antibodies used for Western blotting are listed in Appendix A. Western blots were re-confirmed over 3 biological replicates.

### 2.9. Animal Studies

All procedures involving mice adhered to protocols approved by the Home Office in the United Kingdom under project license PP7510272 held by V.G.B. and were approved by the University of Edinburgh Animal Welfare and Ethical Review Body (PL05-21). Ten-week-old female CD-1/nude mice were intracranially implanted with NPE cells expressing a dual luciferase/GFP reporter (2 × 10^4^) or G7 cells (2 × 10^5^) in 2 µL of growth medium following isoflurane anesthesia. NPE-luciferase/GFP cells were provided from the CRUK Glioma Cellular Genetics Resource and have been previously described [32]. Buprenorphine was administered for analgesia during surgery and carprofen was administered in water for 48 h. Tumor cells were precisely implanted at coordinates 0.6 mm anterior and 1.5 mm lateral to the bregma, reaching a depth of 2.5 mm. Tumor growth was monitored biweekly by injecting 150 mg/kg luciferase, followed by imaging using the IVIS^®^ Lumina S5 system. Mice bearing established tumors were randomized into different treatment groups. The treatment regimen involved administering VS4718 (75 mg/kg) twice daily (7 days a week) and/or trametinib (0.5 mg/kg) once daily (5 days a week) for a total of two weeks via oral gavage. Both the drugs were prepared in 0.5% hydroxypropyl methylcellulose solution. At the end of the treatment, mice were humanely sacrificed through cervical dislocation, and brains were collected, fixed in formalin overnight, and processed for paraffin embedding. Subsequently, sections were cut and subjected to staining with H&E, and anti-GFP (for NPE tumors) and human anti-mitochondrial antibody (for G7 tumors), using conventional techniques.

### 2.10. Immunohistochemistry

Immunohistochemistry was performed using a DAKO (Agilent Technologies, Santa Clara, CA, USA). Briefly, mouse brains were cut into two coronal sections near the implantation sites and embedded in paraffin blocks. Brain slices of 4 µM thickness were placed on glass slides and dewaxed with xylene. Then, 10 mM citrate buffer was used for antigen retrieval in a pressure cooker, and the slides were washed in water and TBS-T. Sections were then blocked with Dako REAL peroxidase block solution (Agilent Technologies, Santa Clara, CA, USA, #S2023) and serum-free protein block solution (Agilent Technologies, Santa Clara, CA, USA, #X0909), followed by overnight incubation with primary antibodies (anti-GFP 1:400) and (human anti-Mitochondria 1:100) (Bio-Techne Ltd., Abingdon, UK, #NBP2-32980). The next day, the sections were washed with TBS-T and incubated with DAKO EnVision-HRP rabbit/mouse (Agilent Technologies, Santa Clara, CA, USA, #K5007) for 2 h. After incubation, the sections were washed again with TBS-T and visualized with DAB reagent (Agilent Technologies, Santa Clara, CA, USA, #K3468). Finally, the sections were counterstained with hematoxylin (Agilent Technologies, Santa Clara, CA, USA, #S3309), dehydrated using ethanol, and mounted using DPX mountant (Merck Life Sciences, Gillingham, UK, #06522).

### 2.11. Reverse-Phase Protein Array

Reverse-Protein Array analysis was performed at the Cancer Research UK Scotland Centre’s Host and Tumour Profiling Unit microarray services. Cell lysates were stored at −70 °C prior to further analysis. The samples were mixed with 4× SDS sample buffer without bromophenol blue and supplemented with 10% 2-β-mercaptoethanol at a final concentration of 2 mg/mL. Samples were denatured by heating to 95 °C for 5 min prior to printing as serial dilutions (1.5, 0.75, 0.375, and 0.1875 mg/mL) in arrays consisting of 36 × 12 spots at a 500 μm spot-to-spot distance on the Aushon 2470 Arrayer Platform (Quanterix, Billerica, MA, USA) using 8 × 185 μM pins, with one deposition round per feature. Each sample dilution series was spotted on all arrays with eight arrays per slide on single-pad ONCYTE SuperNOVA nitrocellulose slides (Grace BioLabs, Bend, OR, USA). Sample loading on the slides for normalization purposes was determined with Fast Green dye staining and scanning using the InnoScan 710 slide scanner (Innopsys, Carbonne, France) at 795 nm. The RPPA slides were then washed with deionized water for 4 × 15 min on a platform shaker and incubated for 10 min with Antigen Retrieval Reagent (1× Reblot strong). The slides were washed twice with deionized water for 5 min. Subsequently, the slides were placed in a ProPlate chamber (Grace BioLabs, OR, USA) in fresh deionized water and washed twice with 1× PBST for 5 min. The slides were then incubated for 10 min in Superblock T20 Blocking Buffer (Pierce, Thermo Fisher Scientific, MA, USA), then washed twice with TBST for 5 min. The RPPA slides were then incubated for 60 min with 127 primary antibodies diluted at 1:250 in Superblock buffer prior to two more TBST washes (5 min each) and a second 10 min incubation with the blocking buffer, as described above. The slides were then incubated for 30 min with the DyLight-800-labelled anti-species secondary antibody diluted at 1:2500 in Superblock, followed by two 5 min washes in TBST, and a final rinse with deionized water. Non-specific background signals were determined for each slide by performing the primary antibody incubation step solely with Superblock, without antibodies, on one array, followed by fluorescent tracer secondary antibody. The slides were dried for 10 min at RT prior to imaging using an InnoScan 710 slide scanner (Innopsys, Carbonne, France). Microarray images were analyzed using Mapix software version 9.1.0 (Innopsys, Carbonne, France). The spot diameter of the grid was set as 270 μm. The background intensity was determined for each spot individually and subtracted from the sample spot signal, thus generating a net signal for each spot. The relative quantification of each analyte was performed for each spot by measuring the fluorescence intensity. The validity of the serial dilutions was ensured by generating a linear fit curve from the 4-point dilution series for all samples on all arrays, using a flag system where an R2 value was generated and R2 >  0.9 values were deemed good. Relative fluorescence intensity (RFI) values corresponding to the relative abundance of total and phosphorylated proteins across the sample set were calculated and normalized to total protein by calculating the ratio of Antibody RFI/Fast-Green RFI for all samples. Fold changes (as Log_2_[Fold]) relative to DMSO controls were calculated and hierarchical clustering (Euclidean distance, complete linkage) was carried out using Morpheus matrix visualization software (https://software.broadinstitute.org/morpheus, accessed on 11 November 2023).

## 3. Results

### 3.1. Chemogenomic Screening to Identify Synergistic Combinations with Loss of FAK Kinase Activity

We sought to identify novel drug combinations that synergize specifically with the loss of FAK in GBM using a chemogenomic screening strategy. We obtained isogenic glioma stem cell models derived from mouse neural stem cells engineered to express constitutively active EGFRvIII and CRIPSR-mediated deletion of *Nf1* and *Pten* [32] (NPE cells). This line was further engineered by CRISPR to express FAK^K454R^ kinase-deficient mutant protein (FAK-KD) and a FAK-re-expression control, representing FAK^wild-type^ protein (FAK-WT) (see Section 2). FAK-WT and FAK-KD lines were adapted to a 384-well format with optimal seeding densities using a cell viability endpoint. Four separate drug libraries were selected, specifically with a drug re-purposing context in mind, and included pre-clinical and/or FDA-approved compounds: the TargetMol anticancer set (L2110, 330 compounds), Kinase Chemogenomic set (KCGS [39], 187 compounds), LOPAC (1280 compounds, Merck), and a custom chemogenomic library set (789 compounds, ‘Comprehensive anti-Cancer small Compound Library, or C^3^L) [34], with dosing across four concentrations (1000, 300, 100, 30 nM). Library dilutions and additions were carried out using a Biomek FX liquid handling robot (Beckman Coulter) with compound exposure held for 48 h and consistent DMSO concentration (0.1% (*v*/*v*)), and cell viability was quantified. Data were normalized to DMSO wells and converted to z-scores for direct FAK-WT vs. FAK-KD comparison (X-Y scatter plots, Figure 1A).

Quality control metrics were assessed on a plate-by-plate basis, specifically robust Z-prime values (based on DMSO vs. staurosporine controls) and normal sample distributions (Appendix A). Compounds that showed increased death in the presence of inactivated FAK (FAK-KD) relative to FAK-WT were assessed by differential analysis, generating a delta-Z score (DZ = WT z-score—KD z-score). Hit compounds were identified and ranked by a DZ score of >2 (Figure 1B), and the complete ranking is presented in Appendix A. The top-ranked hits included previously reported HSP90 inhibitors [40] and several MEK inhibitors, including the clinically advanced drug trametinib. A number of MEK inhibitors were repurchased and validated (Figure 1C), with trametinib (MEK1/2), GDC-0623 (MEK1), cobimetinib (MEK1), and mirdametinib (pan-MEK) selectively inducing cell death in an inactive FAK kinase background relative to FAK-WT cells.

### 3.2. Identifying Synergistic Combinations Between MEK and FAK Inhibitors

A number of FAK inhibitors are currently undergoing clinical development but have demonstrated limited anti-cancer activity when used as single agents, indicating that FAK inhibitors will be most effective when used in combination with other agents [41]. To phenocopy kinase-deficient FAK, we used the well-characterized FAK inhibitor VS4718 to determine if there is synergistic activity in FAK-competent WT cells when used in combination with the MEK inhibitors GDC0632 and trametinib. We dosed FAK-WT 7 × 7 combination matrices. FAK-WT cells were prepared in both 2D and 3D (Matrigel-supported) cultures, followed by dosing and end-point quantification of cell viability. Normalized data (% cell survival) were analyzed for synergy using the SynergyFinder software version 3.0 (synergyfinder.org).

Drug combination data on NPE-FAK-WT cells indicated synergy/additivity when targeting MEK and FAK by chemical inhibition in both 2D and 3D models (Figure 2A–E), using either trametinib or GDC-0623.

A representative heatmap of the FAK+MEK inhibitor combination matrix showing % inhibition (Figure 2A) and the corresponding synergy score and contour map (Figure 2B,C) is shown, where synergy is indicated by maximal or peak synergy scores in darker red (synergy scores ~10+ across four synergy models preferred). Synergy scores across all combination doses in 2D and 3D are shown (Figure 2D), with three synergy models indicated (BLISS, ZIP, Loewe and HSA [35]) [36,37] and the synergistic zone shaded in green. The X-axis indicates the Combination Synergy Score (CSS), which quantifies the potency of the combination (i.e., significant death caused by the synergy observed, indicated by a CSS > 30) [38]. Trametinib combinations with VS4718 in 2D and 3D are indicated by yellow and green data points, and GDC0623 plus VS4718 in 2D and 3D are indicated by blue and red data points. The most synergistic dose combinations that gave combination index (CI) values < 1 are shown in Figure 2E, with the responses in 3D showing greater effects across all four synergy models. Functional and on-target validation of the MEK and FAK inhibitors used in this GBM cell model was carried out, with reduced phosphorylated (p)-FAK and phosphorylated (p)-ERK, as observed by Western blotting (Figure 2F).

### 3.3. FAK+MEK Drug Combination In Vivo: Investigation with Mouse Glioma Model

To evaluate whether the combination of VS4718 and trametinib resulted in greater inhibition of tumor growth in mice, we utilized the NPE model, which is known to form tumors within two weeks following intracranial implantation [32]. Initially, we confirmed drug target engagement by administering three different concentrations of trametinib (0.1, 0.5, and 1 mg/kg), along with a previously established concentration of VS4718 (75 mg/kg). Both drugs were administered via oral gavage for three days. Western blot analysis of tumor lysates showed effective inhibition of p-FAK (Y397) at 75 mg/kg VS4718, and p-ERK (T202/Y204) at 0.5 mg/kg a trametinib (Appendix A).

To assess tumor growth inhibition, we treated NPE-FAK-WT tumor-bearing mice with the individual drugs (VS4718 and trametinib), their combination, and a vehicle control (hydroxypropyl methylcellulose) for two weeks, followed by culling (Figure 3A). Immunohistochemical analysis of the brain sections revealed weak responses to monotherapy, with significant tumor reduction observed only in the combination therapy group (Figure 3B). Quantification of tumor areas based on tumor-specific GFP staining showed a statistically significant reduction in tumor size with combination therapy compared with either monotherapy (Figure 3C). It is important to mention that the combination of higher doses of trametinib (1 mg/kg) and VS4718 (75 mg/kg) likely induced systemic toxicity, as indicated by significant weight loss in the treated mice. In contrast, at a lower dose of trametinib (0.5 mg/kg) combined with VS4718 (75 mg/kg), only one out of five mice exhibited weight loss by the end of the experiment (Appendix A). Our results are consistent with those of similar studies, where the combination of FAK and MEK inhibitors, despite their potent anticancer effects, has been associated with increased toxicity, particularly under more aggressive dosing regimens [42,43].

### 3.4. FAK and MEK Inhibitor Profiling Across Patient-Derived Human Glioma Stem Cells

We further characterized the MEK and FAK inhibitors across a panel of 14 human GBM stem cells (Figure 4). Individual dose–response profiles of MEK inhibitors, trametinib (MEK1/2), GDC-0623 (MEK1), and FAK inhibitor VS4718 were obtained across the human glioma panel. Generally, less than 50% cell growth or death was achieved at up to 1 mM (VS4718) and 300 nM (MEKi) after 72 h exposure (Figure 4A). Additional high content analysis of cell cycle indicates evidence of G1 growth arrest following trametinib treatment of GBM stem cells (Appendix A). Hierarchical clustering, demonstrating drug sensitivity profiles across the GBM stem cell panel for each compound vs. normalized Area Under the Curve (AUC), is shown in Figure 4B.

Representative images of trametinib (75 nM), VS4718 (250 nM) phenotypes versus DMSO (0.1% (*v*/*v*)) are shown for GBM stem cells belonging to the mesenchymal subtype, GCGR-E21 and -E57 cells, indicating increased cell spreading upon trametinib treatment (Figure 4C). We further examined FAK and MEK inhibitor dose combinations across 14 GBM stem cell lines, as 7 × 7 dose combination matrices, to further investigate the FAK+MEK inhibitor synergy across a heterogeneous panel of human GBM stem cells (Figure 5).

Trametinib, GDC-0623, VS4728, and another FAK inhibitor under clinical development, AMP-945 [44], were combined as [FAKi × MEKi] variations and tested across the 14 human GSC lines, in triplicate 7 × 7 matrices (Figure 5E). The data suggest that, while combined targeting of FAK and MEK with small molecules can produce spikes of strong synergy (e.g., E17 cells, Figure 5A–D), in 29% of GBM stem cell lines (4/14) tested in this study, the general trend suggests potent additivity (7/14). Synergy–Additivity–Antagonism zones in Figure 5E (Y-axis, green, grey, red shading) across four synergy models (Bliss, ZIP, HSA, Loewe) are indicated versus the combined sensitivity score (X-axis, CSS, and data points sized by CSS value, >30 required), which effectively quantifies synergy with potency. Across 14 glioma stem cell lines and four synergy models, the synergy vs. CSS scores indicate a range of ‘good-to-potent’ additivity using a FAK+MEK inhibitor combination, with synergy observed in some GBM stem cell lines.

### 3.5. FAK and MEK Inhibitor Combinations Inhibit 3D GBM Spheroid Growth and Invasion

Previous studies have shown that FAK and MEK inhibitors can both inhibit and promote invasion of GBM cells [45,46]. Therefore, we examined the effect of MEK and FAK inhibitor combinations in Matrigel-supported 3D spheroid invasion assays via live cell imaging (Figure 6). Representative images of spheroids at 72 h vs. 264 h are shown (Figure 6A), with quantification of spheroid invasion area (not spheroid size) showing that trametinib alone causes an initial increase in spheroid invasion area due to a more invasive phenotype, as has been reported by other studies [46]. However, time-lapse movies demonstrated that this outgrowth regresses and collapses after ~6 days of exposure to the compound (Figure 6B, green data points) (Appendix A). Similarly, the trametinib-VS4718 drug combination shows an initial increase in spheroid invasion area, which steadily declines after 72 h exposure to drug (Figure 6B, red data points), and spheroid core size shows little growth overall (Figure 6A, phase images: compare far right ‘combination’ panels at 72 vs. 264 h), indicating growth arrest. Overall, our data clearly indicate that low-dose targeting of MEK and FAK inhibitors in combination prevents 3D spheroid growth and invasion in human GBM models.

### 3.6. FAK+MEK Inhibitor Combination Profiling by Reverse-Phase Protein Array (RPPA)

In order to establish tumor signaling pathway effects of the FAK+MEK inhibitor combination, beyond direct targets such as p-ERK, we profiled cell lines either resistant or sensitive to both monotherapies (E13 and E57 cells, respectively) by RPPA (127 verified antibodies, relating to common anti-cancer and adhesion signaling pathways), after treatment with either DMSO 0.1%, trametinib (100 nM), VS4718 (300 nM), or in combination for 24 h. A corresponding experiment was performed in 3D using G7 GBM cells. Sample preparation and RPPA were carried out as described in the Methods section, and differential protein expression analysis across each condition was then carried out relative to DMSO controls, quantified as Log_2_[FOLD] changes (Figure 7A,B).

In E13, E57, and G7 cells, phosphorylated-MEK1/2 (pSer^217/224^) increased relative to the control after trametinib alone (100 nM) and in combination with VS4718 at 24 h (Figure 7A, top row in red plus: pMEK panel). Western blot analysis confirmed the increased phosphorylation of MEK in trametinib-treated cells, while pERK was still efficiently suppressed (Figure 7C–E). This was evident after >6 h of treatment (Appendix A). In monotherapy-resistant E13 cells, the combination showed enhanced suppression of pPYK2^Y402^, pErbB/EGFR^Y1248/1173^, pSTAT5^Y694^, pFLT3^Y591^, pRSK2^S227^, and pJAK2^Y1007/1008^. In monotherapy-sensitive cells, these markers are generally already inhibited, and their combination increases by ~2-fold. Similarly, in G7 cells under 3D culture, there were not many selective fold changes due to the combination alone, with the exception of reduced pRSK2^S227^, which was the only common factor significantly reduced in the combination arm across all three cell lines (Figure 7A,B), along with increased cleaved PARP/apoptosis. Enhanced induction of apoptosis by the FAK+MEK combination was also observed across this panel of cells by Western blotting (Figure 7C–E, panel ‘c-PARP’) and live cell imaging of G7 spheroids with caspase-3 detection reagent (Appendix A).

### 3.7. In Vivo Activity

We further assessed the efficacy of the FAK + MEK inhibitor combination in an orthotopic xenograft model established using human G7 glioblastoma cells implanted intracranially in CD1 nude mice. Following tumor formation, the mice were randomized into four groups and treated with VS4718, trametinib, their combination, or vehicle control (hydroxypropyl methylcellulose) for one week, followed by culling (Figure 8A). Paraffin-embedded brain sections were stained with hematoxylin and eosin and human mitochondrial antibody to visualize and quantify the tumor area. The immunohistochemical images revealed a significant reduction in the tumor area in the combination treatment group compared with either monotherapy (Figure 8B). Positive staining was observed in non-tumor regions of the brain, which was later confirmed as non-specific antibody binding (Appendix A). Quantitative analysis confirmed that the combination of FAK and MEK inhibitors resulted in a statistically significant reduction in tumor size relative to the monotherapy arms, underscoring the potential of this combination therapy in effectively targeting glioblastoma (Figure 8C). However, it should be noted that weight loss, likely due to systemic toxicity, was observed in some mice in the combination treatment group by the end of the one-week treatment period (Appendix A), similar to the findings in the NPE-FAK-WT model.

## 4. Discussion

The objective of our study was to address the unmet clinical need for GBM treatment, focusing specifically on discovering drug combinations that target key effectors in cellular adhesion, invasion, and migration signaling. FAK, a well-known kinase involved in ECM remodeling, has been identified as a therapeutic target in various cancers, including GBM [17,18,19,20,21]. We employed an unbiased chemogenomic screening strategy in our GBM stem cell model system, using isogenic ‘kinase-deficient’ FAK cells (FAK-KD) vs. wild type (FAK-WT) derived from previously established NPE parental cells (NF1^del^, PTEN^del^, EGFRvIII) [32]. We successfully screened four drug repurposing libraries (2586 compounds) across four concentrations (1000–30 nM) yielding a 10,344 paired dataset. Using cell viability endpoints, we identified several inhibitors that demonstrated greater inhibition of survival in FAK-KD cells than in NPE-FAK-WT cells. Overlapping target mechanisms of multiple inhibitors with increased sensitivity in FAK-KD cells included HSP90 [40], MEK/MAP2K, and topoisomerase-I/II inhibitors. Notably, we identified two distinct MEK inhibitors, trametinib and GDC0623, which showed significantly increased sensitivity in FAK-KD cells and became the focus of our subsequent studies (Figure 1).

Our chemogenomic screening suggests that catalytic inhibitors of FAK may phenocopy FAK-KD cells and ‘synergize’ with MEK inhibitors. FAK and MEK inhibitor combinations have previously been reported in uveal melanoma [42,43], RAS-dependent tumors [47], and NF1-related peripheral nerve sheath tumors [48], although clinical efficacy has not been observed in [non-stratified] patients with advanced solid tumors, such as mesothelioma and pancreatic ductal adenocarcinoma [49,50]. Our data from the NPE model suggested that targeting FAK and MEK against an NF1^del^ or EGFRvIII GBM background could act synergistically. We validated this hypothesis using dose-ratio matrix testing of several FAK and MEK inhibitor combinations across multiple 2D and 3D in vitro GBM models. Synergy scores and combination indices <1 confirmed that synergistic activity was mediated through the co-targeting of FAK and MEK (Figure 2A,B). Importantly, we observed additive and synergistic activity at nanomolar concentrations of FAK and MEK inhibitors that are clinically relevant. We further validated the FAK+MEK inhibitor combination in vivo using the murine NPE intracranial transplant model (Figure 3), which showed remarkable antitumor efficacy, resulting in substantial tumor regression. We also validated the FAK+MEK inhibitor combination using the human GBM G7 cell line, which has previously been shown to maintain markers of GBM stem cells and display a growth pattern in CD1 nude mice xenografts characteristic of high-grade gliomas in patients [51] (Figure 8). Established G7 GBM tumors partially responded to monotherapies in vivo; however, in the VS4718+trametinib combination arm, a clear and significant reduction in tumor volume was observed (Figure 8C). Although some systemic toxicity was observed with the combined use of VS4718 and trametinib, the observed synergy suggests dose-sparing potential, supporting the combined targeting of MEK and FAK in GBM as a valid therapeutic approach.

However, the major challenge in GBM is to demonstrate efficacy across heterogeneous tumors. We profiled the FAK inhibitor, VS4718, and the MEK inhibitors, GDC-0623 and trametinib, via dose–response across a panel of 14 patient-derived GBM stem cell models representing broad GBM heterogeneity (Figure 4A). Basic cell survival assays indicated unremarkable potencies as monotherapy, with a spectrum of sensitivity upon treatment with GDC-0623 and trametinib—the latter showing the strongest effects (IC_50_‘s ranging from 80 to 6000 nM) on cells carrying NF1-deletions or EGFRvIII (Figure 4B and Appendix A). Despite this apparent lack of potency, cellular imaging revealed distinct morphological phenotypes upon treatment with FAK and MEK inhibitor monotherapies (representative images in Figure 4C), suggesting distinct mechanisms of action and the potential for combinatorial synthetic lethality to occur. In various [FAKi + MEKi] combinations across our glioma stem cell panel, we quantified cell survival and synergy, describing potency as a ‘combined sensitivity score’ (CSS) vs. synergy–additivity–antagonism scores (Figure 5E). Overall, the FAK+MEK inhibitor combination showed synergistic or potent additive effects across multiple GBM stem cell lines in 2D, and we were able to replicate these drug combination results via live cell imaging in a 3D spheroid invasion assay (Figure 6). Interestingly, in the 3D spheroid invasion assay, trametinib initially appeared to enhance invasion (up to 5 days of drug exposure, day 8), consistent with previous studies that raised concerns that trametinib may promote invasion by strengthening ECM adhesion contacts [46]. Nevertheless, in a 3D spheroid-Caspase-3 assay, only the FAK+MEK inhibitor combination induced apoptosis over 8 days (Appendix A), suggesting that the antitumor cytotoxic activity following long-term treatment with the FAK+MEK combination overcomes any adverse activity of trametinib-induced invasion.

To characterize the signaling pathway profiles in response to the FAK+MEK inhibitor combination, we assessed several cell lines using RPPA under 2D and 3D conditions (Figure 7). A panel of well-characterized antibodies (136 targets, including controls) was assayed against monotherapy-sensitive and monotherapy-resistant cells. Interestingly, in 2D and 3D, the combination appeared to reduce pFAK^Y397^ further (relative to drug alone) and increased pMEK1/2^S217/221^, an apparent feedback signal that was confirmed by time-course studies, but with continued suppression of pERK (Appendix A). The only common factor within the drug combination arms in all three cell models tested here was reduced pRSK2^S227^ and enhanced apoptosis, as demonstrated by Western blotting (cleaved PARP) (Figure 7A,B) and live caspase-3 assays (Appendix A). RSK2, a serine/threonine kinase downstream signaling mediator in the RAS/MEK/ERK pathway, has been identified as driving invasiveness/migration in glioma and other cancers [52,53,54]; hence, it is a favorable downstream target and a potential biomarker of the FAK+MEK inhibitor combination.

Crosstalk and compensatory signaling between the RAS/RAF/MEK and FAK/SRC signaling axes have been reported in multiple cancer cell lines and preclinical tumor models [55,56]. FAK has been shown to be activated following pharmacological inhibition of the RAS/RAF/MEK pathway in several pre-clinical tumor models and patient tumor samples [57,58,59]. FAK signaling can be negatively regulated by the RAS/RAF/MEK pathway through multiple mechanisms. For example, ERK-mediated phosphorylation of FAK on S910 results in FAK inactivation by dephosphorylation and focal adhesion turnover during migration. In addition, ERK-mediated phosphorylation of calpain2 at Ser50 promotes the proteolysis of FAK and focal adhesion disassembly [60,61]. In our study, we observed an increase in the levels of autophosphorylated FAK-Y397 in G7 cells following treatment with trametinib in 3D in vitro cultures (Figure 7B,E). However, this compensatory upregulation of FAK activity was not observed in the other GBM stem cell lines (E57 and E13), which also exhibited enhanced responses to the FAK+MEK combination treatment. In these cell lines, the FAK+MEK combination resulted in enhanced downregulation of FAK-Y397 compared to the FAK inhibitor alone, suggesting that the MEK inhibitors cooperate with the FAK inhibitor to repress upstream pathways promoting focal adhesion signaling.

Importantly, potent additive or synergistic activity of the trametinib + FAK inhibitor combination is observed across multiple murine and human models and is independent of BRAF^V600E^ mutation status or specific compensatory signaling mechanisms, suggesting that this combination may serve a wider population of GBM patients than current personalized treatment propositions, such as the trametinib and dabrafenib combination, which is currently restricted to 1–2% of GBM patients that express the BRAF^V600E^ mutation [13,14].

Combining FAK and RAF/MEK inhibitors has shown clinical activity in low-grade serous ovarian cancer regardless of KRAS mutation status [62] with an ongoing registration-directed study of VS-6766 (RAF/MEK inhibitor) ± defactinib (FAK inhibitor) in patients with recurrent low-grade serous ovarian cancer (ENGOT-ov60/NCRI/GOG-3052; NCT04625270) underway [51]. These studies indicate that the FAK+MEK inhibitor combination may be repurposable for other tumors; however, improving the physicochemical properties of each individual compound component, such as CNS penetrance, may be required to achieve optimal activity in GBM. Improvements in medicinal chemistry, particularly those aimed at improving the CNS penetrance of trametinib—a PgP substrate with poor BBB permeability [63]—are underway. Novel BBB-permeable MEK and ERK inhibitors for BRAF-mutant brain metastases are also being developed [64,65]. Emerging drug delivery devices, such as BBB-penetrating nanoparticles specifically designed for delivering drug combinations [66,67] or focused ultrasound-enhanced delivery [68] to optimize delivery of FAK+MEK inhibitor combinations to GBM tumors, can be considered.

Using the same NPE glioma model used in the current study, we recently reported on another integrin adhesome node, namely, integrin-linked kinase (ILK), which controls transcriptional plasticity [69]. Depletion of ILK drives the conversion of GBM stem cells from a heterogeneous population to a more homogenous one; thus, the FAK+MEK combination treatment may also target plasticity to provide broad activity across heterogeneous GBM stem cell models. Novelty statement: while FAK+MEK drug combinations have been identified to be synergistic in other cancer indications, we believe this is the first study demonstrating potent additive and synergistic activity in GBM models. Here, we provide a comprehensive comparative evaluation of multiple selective FAK and MEK inhibitors tested in combinations across human patient-derived GBM stem cell models and mouse models of GBM. Correlation of FAK+MEK combination activity with molecular characterization of GBM stem cell lines demonstrates no association with the transcriptomic subtype or mutation status of major tumor drivers (NF1, EGFR, B-RAF), indicating this combination may serve a wider population of GBM patients than previously anticipated. Downstream mechanistic analysis reveals a number of pro-apoptosis markers, and reduced phosphorylation of RSK2 on serine position 227 is associated with optimal drug combination response representing potential pharmacodynamic biomarkers. Future studies to promote the translation of FAK+MEK inhibitor combination therapy into GBM clinical trials include broader profiling across a larger panel of diverse GBM stem cell models and correlation of phenotypic outcome with gene expression profiles using advanced statistical analysis methods and computational tools such as VISAGE [70] to identify potential patient stratification biomarkers that predict optimal synergy to support future personalized medicine strategies in GBM. In addition, a full preclinical toxicology package—considering targeted drug delivery strategies such as emerging GBM-targeting nanoparticles [66], if necessary—to decipher safety margins and an optimal therapeutic window of the FAK+MEK inhibitor combination will also build confidence in the clinical potential of FAK+MEK inhibitor combination therapy in GBM.

## Figures and Tables

**Figure 1 pharmaceutics-17-00549-f001:**
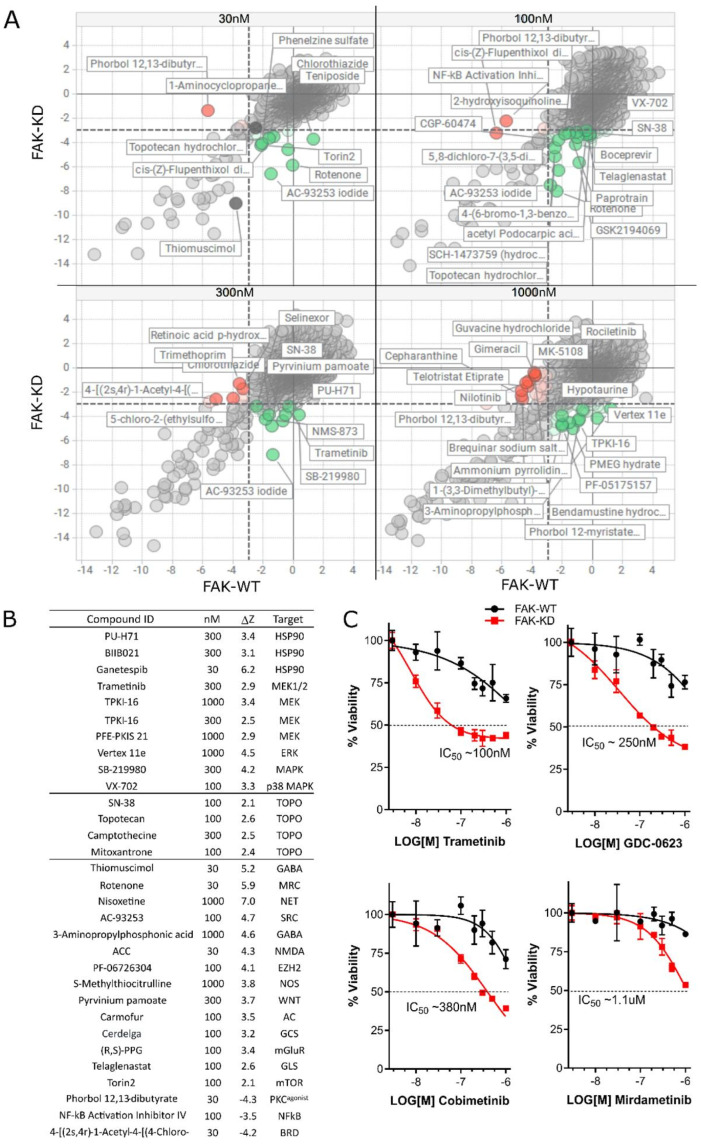
Chemogenomic screening against isogenic FAK-WT and FAK-KD cell lines. (**A**) X-Y Scatter plots of normalized Z-scores of FAK-WT vs. FAK-KD at 30, 100, 300, and 1000 nM. Key compounds are highlighted: FAK-KD sensitizing—green, FAK-KD antagonists—red. (**B**) DZ score rank was generated from thresholds of cell survival of FAK-WT cells (z score > −3) versus increased death in FAK-KD cells (z score < −3). Compound hit list with DZ scores > 2 and higher are listed with their annotated target, plus example antagonistic hits DZ < −3. (**C**) Validation of 4 MEK inhibitors (trametinib, GDC-0623, cobimetinib, mirdametinib) showing loss of FAK kinase activity (red curve) sensitizes cells to MEK inhibition (*n* = 3 biological replicates). The dashed horizontal line indicates 50% cell viability, commonly used to define IC_50_ values.

**Figure 2 pharmaceutics-17-00549-f002:**
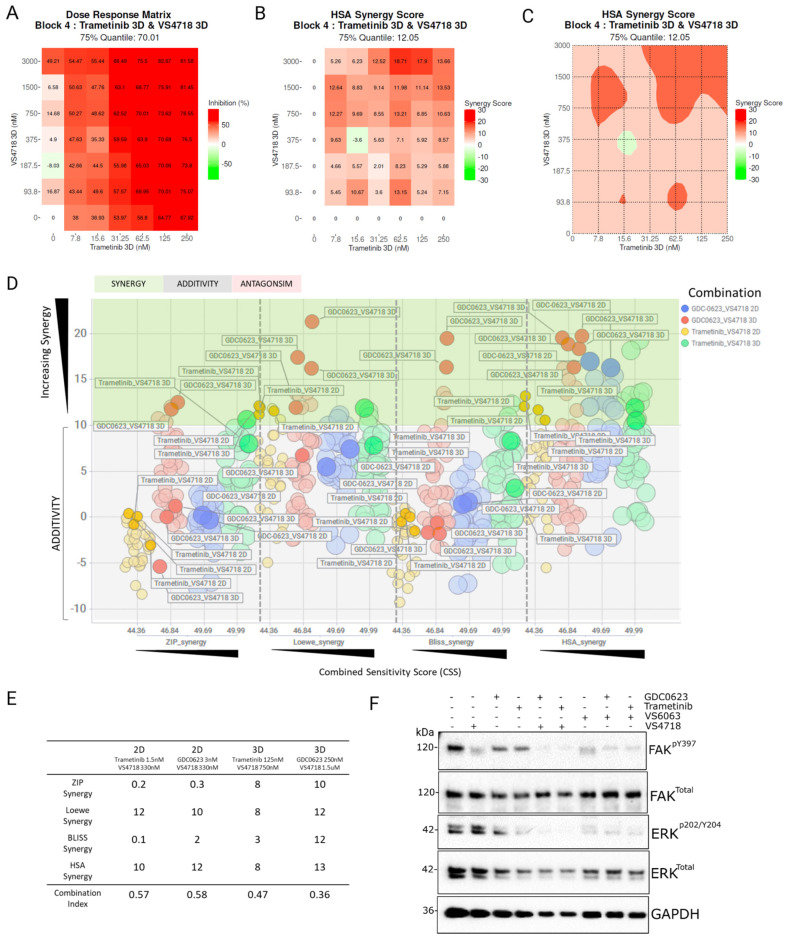
Chemical inhibition of MEK and FAK indicates additive-to-synergistic trends in 2D and 3D NPE-FAK-WT cells. (**A**) Heatmap of % cell survival of combination matrices of VS4718 vs. trametinib in 3D spheroids. (**B**–**C**) Corresponding synergy heatmap and synergy contour plots (HSA model) of trametinib and VS4718 combination matrices, with increasing synergy indicated in red (**D**). Scatter plot of synergy (green zone)/additivity (grey zone) vs. potency (CSS) across 4 synergy models (BLISS, ZIP. Loewe, HSA), from combinations on NPE-FAK-WT cells (2D and 3D) with trametinib or GDC0623 vs. VS4718 (*n* = 3). (**E**) Combination index values indicate improved synergy in 3D vs. 2D models. (**F**) Western blot of validation of on-target trametinib and VS4718 inhibition in this cell model.

**Figure 3 pharmaceutics-17-00549-f003:**
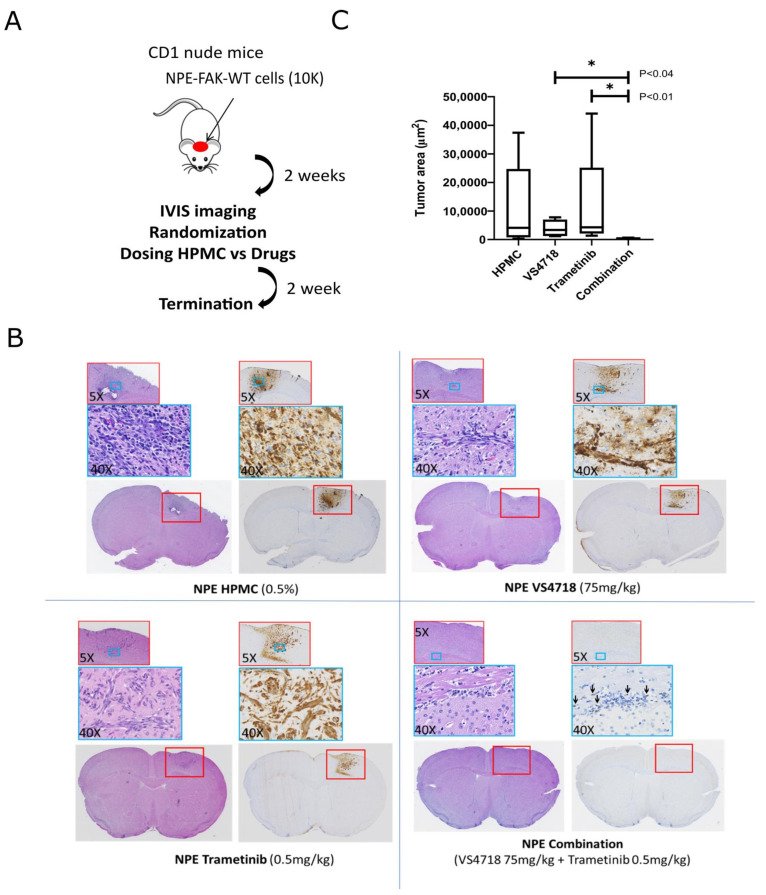
In vivo validation of FAK+MEK inhibitor synergy in NPE FAK-WT tumor-bearing mice. (**A**) Schematic showing experimental protocol where mice with established tumors were treated with drugs (VS4718 and trametinib) or vehicle (hydroxypropyl methylcellulose). (**B**) Representative images of paraffin-embedded brain sections containing NPE-FAK-WT tumors stained with Hematoxylin and eosin or following immunohistochemistry with anti-GFP antibody. Black arrows indicate GFP-stained tumor cells in mouse brain. (**C**) Quantification of Tumor Area based on GFP staining (*n* = 5 each group). Error bars represent the standard error of the mean (SEM). * Indicates statistical significance at *p* < 0.0132.

**Figure 4 pharmaceutics-17-00549-f004:**
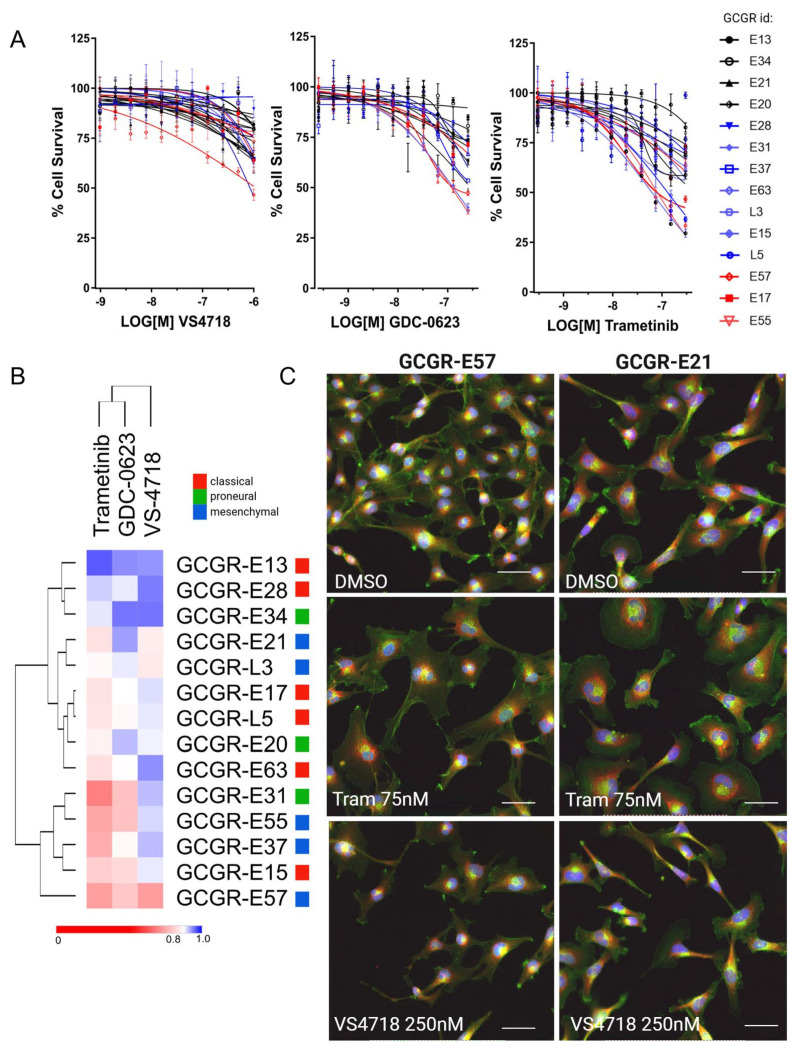
(**A**) Dose–response validation of FAK inhibitor (VS4718) and MEK inhibitors (trametinib and GDC-0623) across 14 patient-derived glioma stem cell (GSC) lines (*n* = 3) (**B**). Heatmap and hierarchical clustering of dose–response data by normalized Area Under the Curve (AUC). AUC data are scaled between 1 (100% inactive) and 0 (100% active). (**C**) Representative images of E57 and E21 cells in response to VS4718 and trametinib at indicated doses (72 h). Staining by Hoechst (Nuclei, blue), Concanavalin A (ER, red), and Phalloidin (F-actin, green). Scale bar indicates 20 µM.

**Figure 5 pharmaceutics-17-00549-f005:**
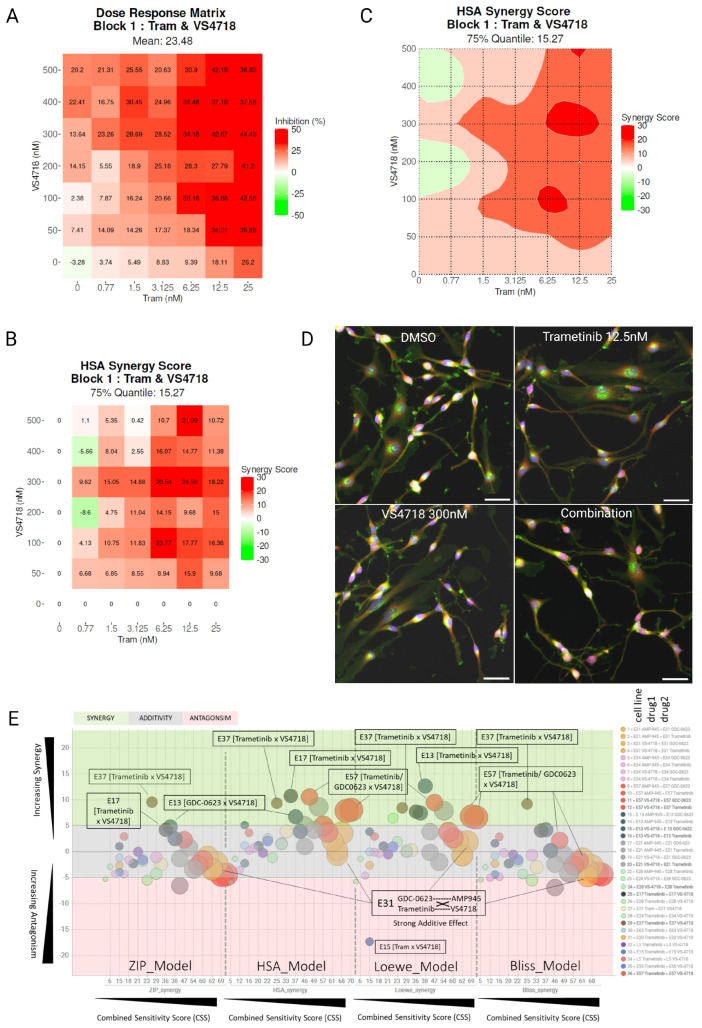
Combination screening (7 × 7 dose combination matrices) of MEK inhibitors (trametinib, GDC-0623) with FAK inhibitors (VS4718, AMP-945) against a panel of 14 patient-derived, human glioma stem cell lines (*n* = 3). (**A**) Representative data: dose combination matrices of % Inhibition (Max Red). (**B**) Synergy combination plot (HSA Model) showing areas of max synergy (red) > 10 dmax and (**C**). Corresponding 2D contour plot showing ideal dose range (dark red zones). (**D**) Representative images of E17 cells treated with DMSO (0.1% *v*/*v*), trametinib (12.5 nM) alone, VS4718 (300 nM) alone and in combination (synergistic doses) staining by Hoechst (Nuclei, blue), Concanavalin A (ER, red), and phalloidin and WGA (F-actin/plasma membrane, green). Scale bar indicates 50 µM. (**E**) Synergy–sensitivity scatter plot of global synergy (across each entire matrix) vs. combined sensitivity score (CSS) of 36 MEKi/FAKi combinations (1296 dose variations) across 14 different cell lines. Sizing by CSS. Some combinations failed QC and were removed. Shading indicates areas of synergy (green), additivity (grey), and antagonism (red). Synergy (>5 synergy score) is calculated across 4 models (Bliss, Loewe, ZIP, HSA) for concurrence.

**Figure 6 pharmaceutics-17-00549-f006:**
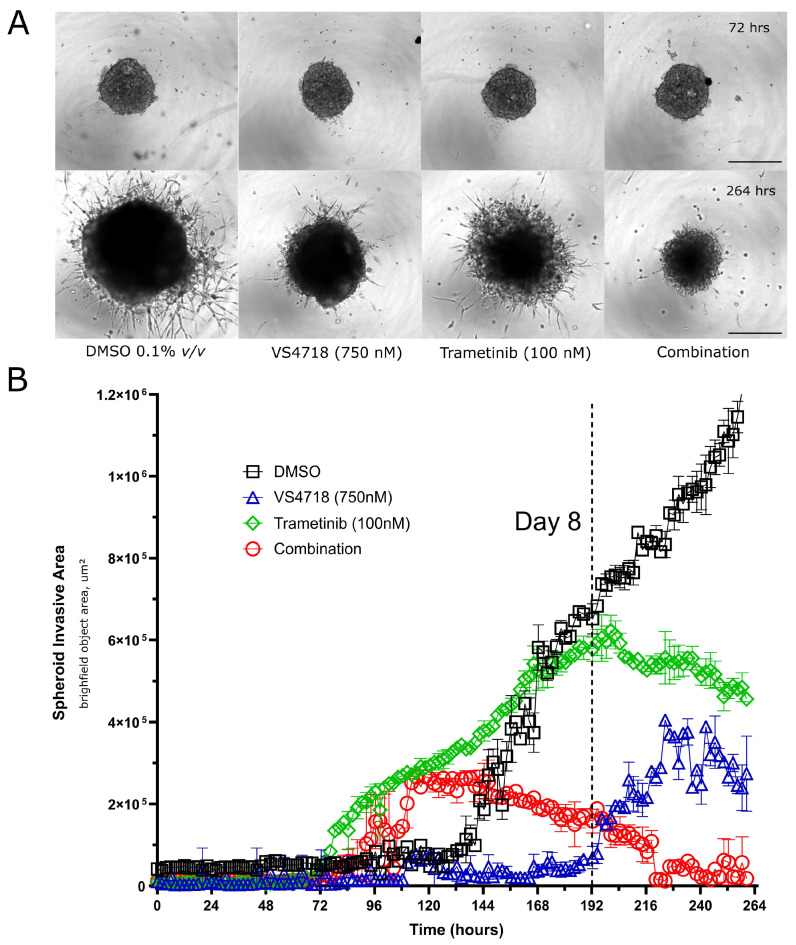
Three-dimensional spheroid invasion assay with FAK and MEK inhibitor combinations. (**A**) Representative images of live cell imaging of spheroids/invasive outgrowth at 72 (‘day 0’) and 264 h (~8 days +drug) after exposure to compounds (indicated concentrations) versus DMSO control (0.1% (*v*/*v*)) (*n* = 3). Scale bar represents 500 mm. (**B**) Quantification of spheroid invasive area (exclusive of spheroid core area) over time (11 days total) (*n* = 3).

**Figure 7 pharmaceutics-17-00549-f007:**
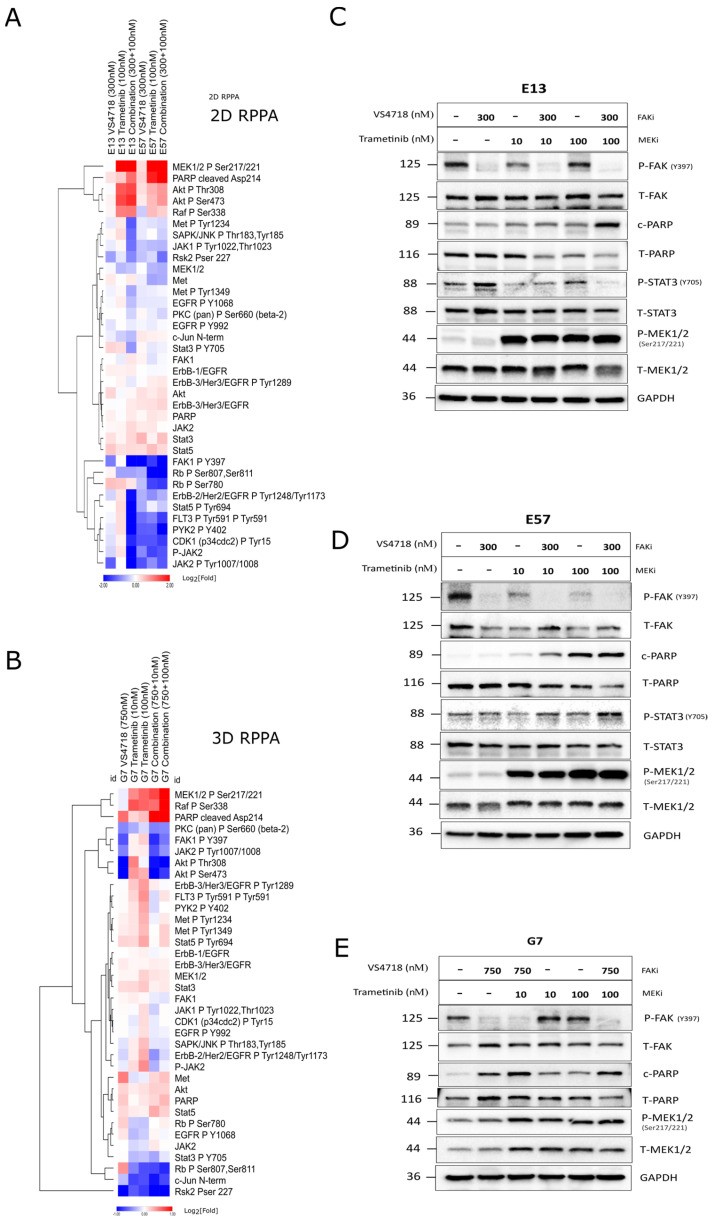
Pathway profiling of FAK+MEK inhibitor combination by Reverse-Phase Protein Array (RPPA) in 2D and 3D. (**A**) Two-dimensional RPPA heatmap representation of protein expression (Log_2_-fold changes) in resistant (E13) vs. sensitivity (E57). (**B**) Three-dimensional RPPA heatmap representation of protein expression (Log_2_-fold changes) in G7 spheroids. (**C**–**E**) Western blot validation of RPPA observations, including enhanced PARP cleavage with FAKi+MEK inhibitor combination (reproduced over a minimum of 3 separate experiments).

**Figure 8 pharmaceutics-17-00549-f008:**
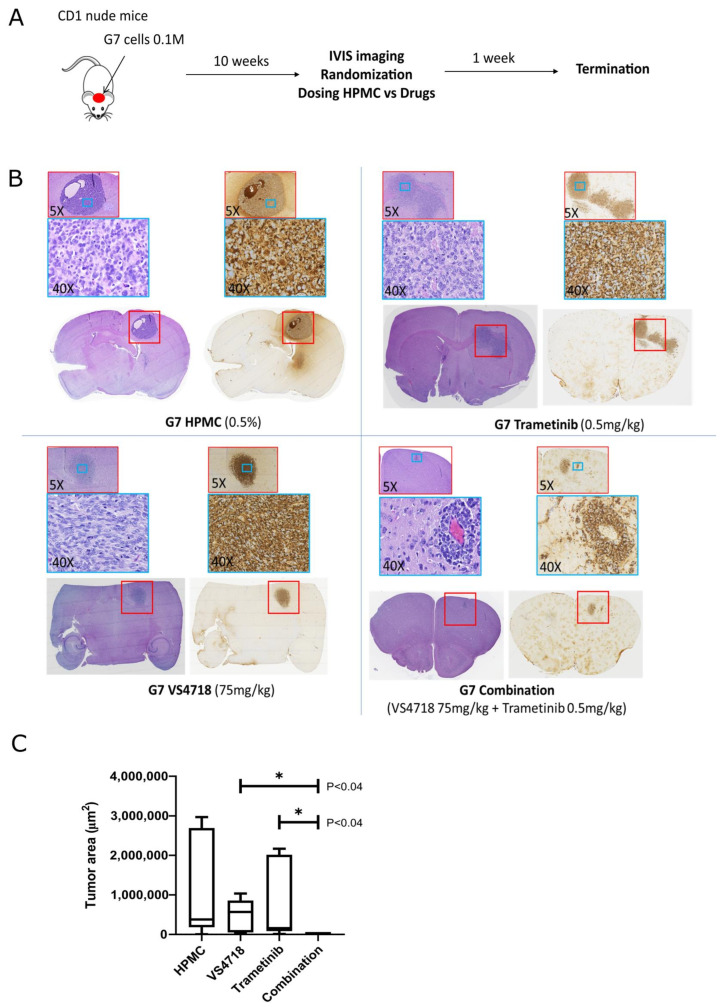
In vivo validation of FAK+MEK inhibitor synergy in G7 tumor bearing mice. (**A**) Schematic showing experimental protocol where mice with established tumors were treated with drugs (VS4718 and trametinib) and vehicle (hydroxypropyl methylcellulose). (**B**) Representative images of paraffin-embedded brain sections containing G7 derived tumors stained with Hematoxylin and eosin or following immunohistochemistry with anti-human mitochondrial antibody. (**C**) Quantification of Tumor Area based on human mitochondrial protein staining (*n* = 5, each group). Error bars represent the standard error of the mean (SEM). * indicates statistical significance at *p* < 0.04.

## Data Availability

All drug screening data files will be made available through the University of Edinburgh data repository: Edinburgh Data Share: https://datashare.ed.ac.uk/ via an open access doi link. Other data and materials will be provided upon request.

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
