# Peer review of "Drug Combinations Targeting FAK and MEK Overcomes Tumor Heterogeneity in Glioblastoma"

_pharmaceutics, 2025, doi:10.3390/pharmaceutics17050549_

Round 1

Reviewer 1 Report

Comments and Suggestions for Authors

The paper by Furqan et al. presents research that support the use of combined MEK and FAK inhibition in GBM treatment. The authors use advanced techniques and GBM models (engineered mouse NPE cells and human GSC cell lines) to achieve their conclusions. The study is of good quality, results are sound, and the combined drug scheme produces very significant results in vivo (it is a bit surprising that the in vivo experiments using mouse NPE modelling GBM were not performed in immunocompetent mice for even higher impact).

However, there are a few points that can be improved to produce a paper of higher quality and clarity for the reader.

Main points:

1) In Fig. 1C, plots represent % cell viability and were obtained using Presto Blue staining. In contrast in Fig. 4A plots represent % cell survival and were obtained by automated nuclei counting. Why were not proper cell viability assays performed for the human cell lines in Fig. 4? The concept of cell viability includes both cell proliferation and cell survival effects. In the experiments showed up to Fig. 4 of the paper the main effect of the drugs seems to be on cell proliferation. This is in agreement with previous studies showing that FAK inhibitors arrest cell proliferation. Effects of possible induction of cell death are presumably shown in Suppl. Fig. 6 (not available to this reviewer) and in the RRPA analysis and subsequent Western blot through activation of c-PARP pathway. It is not clear if drugs induce apoptosis or decrease cell proliferation or both (by the way, were any cell proliferation experiments performed, EdU or similar?). Can the automated system distinguish apoptotic and non-apoptotic nuclei?

2) The title of the paper claims that the new proposed treatment would target GBM heterogeneity. The authors mention that E57 and E21 cell lines are of mesenchymal type, but they do not mention the types of the remaining cell lines tested in their experiments. Therefore, it is difficult to understand if the drug effects are obtained in different cell lines belonging to different GBM subtypes. On the other hand, the NPE model system overexpressing EGFRVIII represents the classical GBM subtype. Whereas drugs affect different human GBM cell lines (Fig. 4), in order to claim the overcoming of GBM heterogeneity it would be necessary to emphasize the subtypes of GBM cell lines used and also to include this aspect in the Discussion.     

3) Please indicate in Methods and/or legends the number of replicates for all the experiments.

4) Supplementary Fig. 6 (apoptosis analysis) and 4 are not available to this reviewer.

Minor points to correct:

  • 3: GFP is used as a reporter of the implanted cells. There is no reference to GFP expression in NPE cells in Methods.
  • No references in the Abstract.
  • Please correct punctuation errors (lane 83 coma, lane 79 secondary in capitals, legend of Fig 3 double dot, etc.) and typos (“In order To” , overcomes in the title, etc.).
  • Reference #33 is duplicated (ref. 48).
  • Excessive use of abbreviations: avoid or develop (for example, HMCS, LGSOC..)
  • Molecular weight markers should be indicated in the Supplementary Material (uncrop Western blots). Add “kDa” in Figure 2 and 7.

Author Response

Reviewer 1

The paper by Furqan et al. presents research that support the use of combined MEK and FAK inhibition in GBM treatment. The authors use advanced techniques and GBM models (engineered mouse NPE cells and human GSC cell lines) to achieve their conclusions. The study is of good quality, results are sound, and the combined drug scheme produces very significant results in vivo (it is a bit surprising that the in vivo experiments using mouse NPE modelling GBM were not performed in immunocompetent mice for even higher impact).

However, there are a few points that can be improved to produce a paper of higher quality and clarity for the reader.

Main points:

1) In Fig. 1C, plots represent % cell viability and were obtained using Presto Blue staining. In contrast in Fig. 4A plots represent % cell survival and were obtained by automated nuclei counting. Why were not proper cell viability assays performed for the human cell lines in Fig. 4? The concept of cell viability includes both cell proliferation and cell survival effects. In the experiments showed up to Fig. 4 of the paper the main effect of the drugs seems to be on cell proliferation. This is in agreement with previous studies showing that FAK inhibitors arrest cell proliferation. Effects of possible induction of cell death are presumably shown in Suppl. Fig. 6 (not available to this reviewer) and in the RRPA analysis and subsequent Western blot through activation of c-PARP pathway. It is not clear if drugs induce apoptosis or decrease cell proliferation or both (by the way, were any cell proliferation experiments performed, EdU or similar?). Can the automated system distinguish apoptotic and non-apoptotic nuclei?

AU: We thank the reviewer for their constructive comments on our manuscript. Our preferred method for quantifying phenotypic response of cells following drug and drug combination treatments is automated high content imaging, this technology provides more mechanistic information including cell survival, cell cycle status and other morphological effects such as senescence which can all be determined from a single nuclei stain. Also, a limitation of most viability assays (e.g. MTT / AlamarBlue / PrestoBlue) is that they do not provide direct measurement of cell proliferation or survival but rather quantify metabolic activity required to reduce resazurin-based reagents to a more fluorescent compound. Thus such viability assays present a risk of picking up false positive hit compounds which potently and selectively target metabolic pathways with less potent effects on cell survival and proliferation. In response to the reviewers question it is possible to distinguish apoptotic and non-apoptotic nuclei for automated high content image analysis as apoptotic nuclei stained with DNA-binding dyes such as Hoechst appear brighter and smaller than their non-apoptotic counterparts, however such analysis is often an underestimation of apoptosis as many apoptotic cells will have detached from the cell substrates and will not be included in such analysis. Thus we believe a more robust assay for measuring apoptosis is the use of cell-permeable caspase biosensor fluorescent probes that can be loaded into cells and monitored kinetically following drug treatment as presented in our original supplementary figure 6 copied below.

We do agree with the reviewer that the results presented indicate that the FAK and MEK inhibitors are inhibiting cell proliferation. We now confirm this by including additional image analysis using the MetaXpress cell cycle analysis module, which is integrated within the ImageXpress high content imaging system. This analysis classifies each Hoechst stained nuclei into different stages of the cell cycle based on total DNA content.

Through this analysis we do see G0/G1 arrest in ‘drug-alone’ arms – particularly with the more potent compound, trametinib (new Supplementary Figure 3 added to our revised manuscript and copied below). In E21 cells, this G0/G1 arrest is significantly enhanced with the VS4718/trametinib combination relative to trametinib alone (compare Trametinib 25nM [red bar] vs Combi 25-500 [green bar], Supplementary Figure 3A). In E17 cells, this G0/G1 arrest is not significantly enhanced with the VS4718/trametinib combination (Supplementary Figure 3B, ie cells that are more resistant to the FAK inhibitor, VS-4718), nevertheless sustained G0/G1 arrest results in induction of apoptosis, as seen via cleaved PARP (RPPA,  western blotting, Figure 7 in E13, E57) and selective activation of caspase-3 in 3D spheroids observed only in the drug combination arm treatments and not with FAK or MEK inhibitor monotherapy (live cell imaging, Original Supplementary Figure 6 (now supplementary figure 7 – copied below). We highlight these results demonstrating inhibition of GBM stem cell proliferation with monotherapy (Line 448 revised manuscript).

 Supplementary figure 3

Supplementary Figure 3:  A. Cell cycle analysis of E21  and B. E17 cells [from Hoechst-33342 nuclear staining, DNA content analysis, (MetaXpress Software). Statistical analysis (Two-Way Anova) was carried out using  GraphPad Prism.

Original supplementary figure6 

In summary, we believe this additional analysis clearly indicates that FAK and MEK inhibitors can suppress GBM stem cell proliferation when used as monotherapy, but only in combination do they induce apoptosis.

To note: The isogenic mouse NPE-FAK WT and KD cell lines used for the primary screening study in Figure 1 are only partly adherent and thus not suitable for image-based quantification.

2) The title of the paper claims that the new proposed treatment would target GBM heterogeneity. The authors mention that E57 and E21 cell lines are of mesenchymal type, but they do not mention the types of the remaining cell lines tested in their experiments. Therefore, it is difficult to understand if the drug effects are obtained in different cell lines belonging to different GBM subtypes. On the other hand, the NPE model system overexpressing EGFRVIII represents the classical GBM subtype. Whereas drugs affect different human GBM cell lines (Fig. 4), in order to claim the overcoming of GBM heterogeneity it would be necessary to emphasize the subtypes of GBM cell lines used and also to include this aspect in the Discussion.    

AU: We thank the reviewer for highlighting this omission, we have now added GBM subtype designation (Mesenchymal, Classical, Proneural) as determined by transcriptomic analysis to each GBM cell line (colour coded) presented in Figure 4 of the revised manuscript. In addition, we provide information on transcriptomic subtype designation and copy number and mutational status of EGFR, NF1 and IDH for each GBM cell line in supplementary Table 1 (copied below) and now refer to this information in our discussion of the revised manuscript (line 740-742).

Supplementary table1

Supplementary Table 1. Molecular classification of patient-derived GBM stem cell lines.

3) Please indicate in Methods and/or legends the number of replicates for all the experiments.

AU: We now provide this information for each experiment throughout the methods section and figure legends of our revised manuscript.

4) Supplementary Fig. 6 (apoptosis analysis) and 4 are not available to this reviewer.

Au: Supplementary figures have been submitted for review. In addition we have provided original supplementary Figure 6 above and supplementary Figure 4 below [now supplementary Fig 7 and supplementary Fig 5 in the revised manuscript]:

 Supplementary figure4A

supplementary figure 4B

Supplementary Figure 4 (now Supp Figure 5): Body weight measurements of NPE-FAK-WT and G7 tumour bearing mice treated with the indicated drugs over the course of two and one week, respectively via oral gavage. Relative weight was calculated by dividing the weight of each mouse on a given day by its weight on day 1. Each coloured line represents one mouse.

Minor points to correct:

3: GFP is used as a reporter of the implanted cells. There is no reference to GFP expression in NPE cells in Methods.

AU: The NPE-GFP cells used in this study were provided by the Glioma Cellular and Genetics Resource and have previously been described in Gangoso E, et al., Cell. 2021 Apr 29;184(9):2454-2470.e26. doi: 10.1016/j.cell.2021.03.023. We now provide this information in our methods section (line 276-279 of the revised manuscript).

No references in the Abstract.

Au: We have now removed the reference from the abstract.

Please correct punctuation errors (lane 83 coma, lane 79 secondary in capitals, legend of Fig 3 double dot, etc.) and typos (“In order To” , overcomes in the title, etc.).

Au: We have now corrected.

Reference #33 is duplicated (ref. 48).

Au: We have removed the duplicated reference

Excessive use of abbreviations: avoid or develop (for example, HMCS, LGSOC..)

Au: We have removed or replaced many of the abbreviations use throughout the manuscript including LGSOC, HPMC etc.

Molecular weight markers should be indicated in the Supplementary Material (uncrop Western blots). Add “kDa” in Figure 2 and 7.

Au: All uncropped western blots, including molecular weight markers, presented in the manuscript and supplementary figures have been included  as raw data files along with our submission as per editorial instructions.  We now also add “kDa” in Figs 2 and 7 and supplementary Figs 1,4 and 6. 

Reviewer 2 Report

Comments and Suggestions for Authors

The manuscript presents a well-structured and comprehensive study investigating the synergistic potential of FAK and MEK inhibitors for glioblastoma (GBM). The research incorporates in vitro, in vivo, and pathway profiling experiments, making it a valuable contribution to the field. However, several key areas require further attention before the manuscript can be considered for publication. One primary concern is the high similarity index (85%) in iThenticate, which raises questions about originality. Additionally, the manuscript would benefit from a clearer justification of its novelty, enhanced clinical relevance, a more thorough toxicity assessment, and a discussion on blood-brain barrier (BBB) permeability to strengthen its impact.

Major Concerns:

  1. Plagiarism Concern (85% iThenticate Similarity Score) Must Be Addressed
    The high similarity index suggests significant textual overlap, which needs careful revision. The manuscript should be rewritten where necessary to ensure originality while maintaining scientific accuracy. If self-plagiarism exists, previously published work should be properly cited to provide transparency. A revised version should aim for a significantly lower similarity score (≤20%).
  2. While this study explores FAK and MEK inhibitors, similar strategies have been investigated previously. To enhance its contribution, the manuscript should emphasize what distinguishes this study. If a new mechanism, biomarker, or improved therapeutic effect has been identified, this should be clearly stated. A clearly defined novelty statement will ensure that the study is recognized as an advancement rather than merely a confirmatory analysis.
  3. The study effectively uses patient-derived glioblastoma stem cells (GSCs), which is commendable. However, validation using clinical biopsy data or survival correlations is currently missing. To enhance translational relevance, the authors may consider aligning their findings with patient datasets (e.g., TCGA), proposing biomarker-based patient selection, and comparing their approach with existing GBM treatments. This would provide a stronger foundation for potential clinical application.
  4. The study reports weight loss in treated animals, which suggests potential systemic toxicity. However, a detailed toxicity and pharmacokinetics (PK) evaluation is not provided. Incorporating hematological, biochemical, and histopathological analyses would help determine drug metabolism, clearance rates, and overall safety. If toxicity is observed, exploring alternative delivery methods (e.g., nanoparticles) could be a valuable consideration to improve therapeutic outcomes.
  5. Given that BBB penetration is a critical challenge in GBM therapy, it would be helpful if the manuscript addressed whether the FAK and MEK inhibitors effectively reach the tumor site. If available, existing data on CNS penetration should be included. Potential solutions such as nanoparticles or focused ultrasound-enhanced delivery could be discussed if BBB permeability is a concern. This would provide a more comprehensive perspective on the feasibility of this therapeutic strategy.
  6. The manuscript concludes that FAK and MEK inhibition is a promising strategy for GBM; however, it does not specify which patient subgroups are most likely to benefit. Given the heterogeneity of GBM, not all patients may respond equally to this treatment. Identifying predictive biomarkers (e.g., EGFR mutations, NF1 deletions, RSK2 expression) could enhance the study's clinical relevance and translational potential. The conclusion should be more precise, avoiding broad claims while acknowledging the need for patient stratification in future studies.

Author Response

Reviewer 2

Comments and Suggestions for Authors

The manuscript presents a well-structured and comprehensive study investigating the synergistic potential of FAK and MEK inhibitors for glioblastoma (GBM). The research incorporates in vitro, in vivo, and pathway profiling experiments, making it a valuable contribution to the field. However, several key areas require further attention before the manuscript can be considered for publication. One primary concern is the high similarity index (85%) in iThenticate, which raises questions about originality. Additionally, the manuscript would benefit from a clearer justification of its novelty, enhanced clinical relevance, a more thorough toxicity assessment, and a discussion on blood-brain barrier (BBB) permeability to strengthen its impact.

Major Concerns:

Plagiarism Concern (85% iThenticate Similarity Score) Must Be Addressed

The high similarity index suggests significant textual overlap, which needs careful revision. The manuscript should be rewritten where necessary to ensure originality while maintaining scientific accuracy. If self-plagiarism exists, previously published work should be properly cited to provide transparency. A revised version should aim for a significantly lower similarity score (≤20%).

Au: We think it would be very strange for an original research article like ours to have such a high 85% similarity score with a separate publication. In response to the reviewers concern we ran our article through the IThenticate Plagirism Detection Software and the three hits with >85% similarity all link to our pre-print article of this same study that we posted on bioRxiv on December 1st 2024: https://www.biorxiv.org/content/10.1101/2024.11.26.625442v1.full.pdf. The next most similar article has a 5% iThenticate similarity score and refers to a separate GBM manuscript posted by our research group where text similarity relates to authors names, affiliations, conflict statements and funding acknowledgements. We are therefore confident that our manuscript represents an original research article with no evidence of plagiarism or self-plagiarism.

While this study explores FAK and MEK inhibitors, similar strategies have been investigated previously. To enhance its contribution, the manuscript should emphasize what distinguishes this study. If a new mechanism, biomarker, or improved therapeutic effect has been identified, this should be clearly stated. A clearly defined novelty statement will ensure that the study is recognized as an advancement rather than merely a confirmatory analysis.

Au: We thank the reviewer for this helpful suggestion. We now provide a novelty statement towards the end of our discussion of the revised manuscript (line 735-746). This relates to the following novel aspects of our manuscript:

  • While, FAK+MEK drug combinations have been identified to be synergistic in other cancer indications we believe this is the first study demonstrating potent additive and synergistic activity in glioblastoma (GBM) models.
  • Importantly we provide the first comprehensive study robustly evaluating and comparing multiple selective FAK and MEK inhibitors in combinations across human patient-derived GBM stem cell models and mouse models of GBM.
  • Correlation of FAK+MEK combination activity with molecular characterization of GBM stem cell lines demonstrates no association between optimal drug combination response with transcriptomic subtype or copy number/mutation status of major tumour drivers (NF1, EGFR, B-RAF(all of our cell lines are wild-type for B-Raf) indicating this combination may serve a wider population of GBM patients than previously anticipated.
  • Downstream mechanistic analysis, including phosphoproteomics reveals a number of pro-apoptosis markers and reduced phosphorylation of RSK2 on serine position 227 is associated with drug combination response representing potential pharmacodynamic biomarkers for this combination.

The study effectively uses patient-derived glioblastoma stem cells (GSCs), which is commendable. However, validation using clinical biopsy data or survival correlations is currently missing. To enhance translational relevance, the authors may consider aligning their findings with patient datasets (e.g., TCGA), proposing biomarker-based patient selection, and comparing their approach with existing GBM treatments. This would provide a stronger foundation for potential clinical application.

Au: While we agree with the reviewer that testing the FAK+MEK drug combination on patient biopsies could provide useful information with regards to translational potential we believe such experiments are challenging for robust testing of drug combinations. These challenges include access to a fresh supply of patient biopsies which is currently beyond the scope of our group. In addition, limited throughput of such assays limit robust testing of multiple drug combinations across multiple dose ranges to formerly calculate additive, synergistic or antagonistic response. We are unsure whether such biopsy assays provide suitable reproducibility to accurately quantify and compare drug potency EC50 values and drug combination activity across different patient samples. With regards to bioinformatics analysis, currently there is not publicly available data on GBM patient samples treated with selective FAK or MEK inhibitors thus we are not sure how we can use bioinformatics to align our findings with current patient datasets such as TCGA to provide useful information.

The study reports weight loss in treated animals, which suggests potential systemic toxicity. However, a detailed toxicity and pharmacokinetics (PK) evaluation is not provided. Incorporating hematological, biochemical, and histopathological analyses would help determine drug metabolism, clearance rates, and overall safety. If toxicity is observed, exploring alternative delivery methods (e.g., nanoparticles) could be a valuable consideration to improve therapeutic outcomes.

Au: A full preclinical toxicology package on FAK + MEK drug combination is beyond the capabilities of our academic group. However, in response to the reviewers comments we do now suggest these studies should be carried out with a view to exploring alternative delivery methods such as recently developed GBM targeting nanoparticle strategies in the discussion of our revised manuscript (line 752-754 revised manuscript) .

Given that BBB penetration is a critical challenge in GBM therapy, it would be helpful if the manuscript addressed whether the FAK and MEK inhibitors effectively reach the tumor site. If available, existing data on CNS penetration should be included. Potential solutions such as nanoparticles or focused ultrasound-enhanced delivery could be discussed if BBB permeability is a concern. This would provide a more comprehensive perspective on the feasibility of this therapeutic strategy.

Au: While clinical studies demonstrate evidence of blood-brain barrier (BBB) permeability and efficacy for trametinib in the treatment of CNS tumours such as glioblastoma [Kushnirsky M, et al. Prolonged Complete Response With Combined Dabrafenib and Trametinib After BRAF Inhibitor Failure in BRAF-Mutant Glioblastoma. JCO Precis Oncol. 2020 Feb 3;4:PO.19.00272. doi: 10.1200/PO.19.00272]and ganglioglioma – these studies also indicate that trametinib is a substrate for the P-glycoprotein (P-gp) efflux transporters limiting brain distribution [Vaidhyanathan S, et al.,. Factors influencing the CNS distribution of a novel MEK-1/2 inhibitor: implications for combination therapy for melanoma brain metastases. Drug Metab Dispos. 2014 Aug;42(8):1292-300. doi: 10.1124/dmd.114.058339]. We could not find any published data on CNS penetration of the FAK inhibitors used in our study, however another FAK inhibitor GSK2256098 has been reported to cross the BBB at low levels into normal brain, but at markedly higher levels into glioblastoma tumour [Brown NF, et al. A study of the focal adhesion kinase inhibitor GSK2256098 in patients with recurrent glioblastoma with evaluation of tumor penetration of [11C]GSK2256098. Neuro Oncol. 2018 Nov 12;20(12):1634-1642. doi: 10.1093/neuonc/noy078.]. Although we observe efficacious responses in our intracranial GBM transplant in vivo mouse studies with both FAK (VS41718) and MEK (trametinib) inhibitors it remains unclear precisely how much free drug fraction of either FAK or MEK inhibitors accumulates in the human brain to sustain simultaneous target inhibition of FAK and MEK. As the reviewer suggests we add additional comment in our discussion on the potential of emerging BBB penetrating nanoparticles or focused ultrasound-enhanced delivery to optimize drug combination delivery to brain tumours (line 726-729 revised manuscript).

The manuscript concludes that FAK and MEK inhibition is a promising strategy for GBM; however, it does not specify which patient subgroups are most likely to benefit. Given the heterogeneity of GBM, not all patients may respond equally to this treatment. Identifying predictive biomarkers (e.g., EGFR mutations, NF1 deletions, RSK2 expression) could enhance the study's clinical relevance and translational potential. The conclusion should be more precise, avoiding broad claims while acknowledging the need for patient stratification in future studies.

Au: Our study clearly indicates no correlation between FAK+MEK combination response with transcriptomic subtype, copy number or mutation status of major tumour drivers (e.g. NF1, EGFR) indicating this combination may serve a broad population of GBM patients. However, we acknowledge the reviewer’s comments and in our discussion  we suggest a broader panel screen across a larger collection of molecularly annotated GBM stem cell lines followed by established statistical and computational analysis (e.g. VISAGE: DOI:10.1016/j.cels.2019.05.009) to identify potential patient stratification biomarkers for drug combinations and guide future personalized medicine clinical trials (line 746-752).

Reviewer 3 Report

Comments and Suggestions for Authors

Comments

The study of Furqan M. et al, is a well-designed and performed study describing the combination of FAK and MEK inhibitors in the treatment of GBM. The work is timely and very important, since there are no effective treatment for brain tumors. Overall, most of the work is well written, introduction is clear, materials and methods have enough details, results are clear and discussion/conclusion well supported by the results. The reviewer supports the publication of the manuscript after minor changes:

Line 29 – Authors mentioned trametinib but not the FAK inhibitor

Line 39 – Why not mentioning MEK inhibitors as well?

Line 49 – tumor microenvironment (TME)

Line 79 and 83 – Punctuation errors

Line 85 – Rephrase

Line 151 – describe EGFR

Author Response

Reviewer 3

The study of Furqan M. et al, is a well-designed and performed study describing the combination of FAK and MEK inhibitors in the treatment of GBM. The work is timely and very important, since there are no effective treatment for brain tumors. Overall, most of the work is well written, introduction is clear, materials and methods have enough details, results are clear and discussion/conclusion well supported by the results. The reviewer supports the publication of the manuscript after minor changes:

Au: We thank the reviewer for their positive comments

Line 29 – Authors mentioned trametinib but not the FAK inhibitor

Au: The main FAK inhibitor we have studied (VS4718 ) is now mentioned

Line 39 – Why not mentioning MEK inhibitors as well?

Au: MEK now included

Line 49 – tumor microenvironment (TME)

Au: Corrected

Line 79 and 83 – Punctuation errors

Au: Corrected

Line 85 – Rephrase

Au: We have rephrased as follows: “ While only 2% of adult GBM express BRAF V600E, the incidence of activating oncogenic BRAFV600E mutation is higher across other central nervous system primary tumors in pediatric and adult patients [15], thus Dabrafenib combined with trametinib represents a potential treatment option for these subsets of CNS tumour patients. “

Line 151 – describe EGFR

Au: Now described fully (line 165 of the revised manuscript)

Round 2

Reviewer 2 Report

Comments and Suggestions for Authors

The authors have satisfactorily addressed all the concerns raised in the initial review. The revised manuscript has been significantly improved and is now suitable for publication in Pharmaceutics in its current form.